# Tensile Strength and Moisture Absorption of Sugar Palm-Polyvinyl Butyral Laminated Composites

**DOI:** 10.3390/polym12091923

**Published:** 2020-08-26

**Authors:** Shamsudin N. Syaqira S, Z. Leman, S. M. Sapuan, T. T. Dele-Afolabi, M. A. Azmah Hanim, Budati S.

**Affiliations:** 1Department of Mechanical and Manufacturing Engineering, Faculty of Engineering, Universiti Putra Malaysia, 43400 Serdang, Selangor, Malaysia; syaqirashamsudin@gmail.com (S.N.S.S.); sapuan@upm.edu.my (S.M.S.); deleafolabitemitope@gmail.com (T.T.D.-A.); azmah@upm.edu.my (M.A.A.H.); sindhubudati408@gmail.com (B.S.); 2Laboratory of Biocomposite Technology, Institute of Tropical Forestry and Forest Products (INTROP), Universiti Putra Malaysia, 43400 Serdang, Selangor, Malaysia; 3Faculty of Engineering, Universiti Putra Malaysia, Advanced Engineering Materials and Composites Research Center, (AEMC), 43400 Serdang, Selangor, Malaysia

**Keywords:** laminate composites, polyvinyl butyral (PVB), sugar palm fiber (SPF), tensile strength, moisture absorption

## Abstract

Natural fiber reinforced composites have had a great impact on the development of eco-friendly industrial products for several engineering applications. Sugar palm fiber (SPF) is one of the newly found natural fibers with limited experimental investigation. In the present work, sugar palm fiber was employed as the natural fiber reinforcement. The composites were hot compressed with polyvinyl butyral (PVB) to form the structure of laminated composites and then were subjected to tensile testing and moisture absorption. The maximum modulus and tensile strength of 0.84 MPa and 1.59 MPa were registered for samples PVB 80-S and PVB 70-S, respectively. Subsequently, the latter exhibited the highest tensile strain at a maximum load of 356.91%. The moisture absorption test revealed that the samples exhibited better water resistance as the proportion of PVB increased relative to the proportion of SPF due to the remarkable hydrophobic property of PVB in comparison with that of SPF.

## 1. Introduction

Natural fiber reinforced polymer composites are composite materials composed of a polymer matrix imbedded with natural fiber reinforcement. Recent investigations have showcased natural fiber reinforced composites as viable alternatives to glass reinforced composites for the development of both structural and functional components, especially in the automotive, aircraft, and architectural industries. Natural fibers exhibit exceptional properties such as light weight, low density, low cost, comparable tensile qualities, non-abrasiveness with respect to equipment, non-irritating with respect to skin, renewability, reduced energy consumption, recyclable, and biodegradable [1]. However, this composite system is marred by drawbacks such as sensitivity to water and moisture absorption, its polar and hydrophilic nature, quality variations, and low thermal stability [2].

Sugar palm tree can produce various kinds of products like palm sugar, fruits, and fibers [3]. Sugar palm tree is a species that can be found naturally in forests, and it was originally a member of the Palmae family and belongs to the subfamily of Arecoideae and tribe Caryoteae [4,5,6]. Sugar palm is also known to be a fast growing palm, which can achieve maturity within 10 years [7]. The geographical distribution of sugar palm covers the Indo-Malay region and as wide as South Asia to South East Asia [8]. Moreover, by using this fiber, the plant waste can be utilized. In addition, the sugar palm fiber can easily be found, as it highly available at a very low cost. Bachtiar et al. investigated the tensile properties of sugar palm fiber and reported values of a 3.69 GPa Young’s modulus, 190.29 MPa tensile strength, 19.6% strain to failure, and density of 1.26 kg/m^3^ [9].

Polyvinyl butyral (PVB) is a polyacetal structure produced by the process of the condensation of polyvinyl alcohol with n-butyraldehyde in the presence of an acid catalyst [10]. Previous research had shown PVB to be a random terpolymer that contains butyral and hydroxyl side groups with a small amount of acetate units under ^13^ C NMR spectroscopy [11]. The final structure can be a random terpolymer of 76% vinyl butyral, 22% vinyl alcohol, and 2% vinyl acetate [12]. PVB has excellent ductility and adhesive properties with respect to the glass plane. The value of the strain rate in the low-speed tensile test for PVB is 0.008 s^−1^–0.317 s^−1^, while the value of the strain rate in the high-speed tensile test is 8.7 s^−1^–1360 s^−1^. PVB also behaves as a hyperelastic material influenced by the loading rate, while the response of PVB is characterized by a time-dependent non-linear elastic behavior under dynamic loading. The ductility of PVB reduces as the strain rate increases. Thus, it has been proposed that PVB has outstanding mechanical properties and excellent optical clarity [13]. The stress-strain curves of PVB under high strain rates are elastoplastic, whereas, at low strain rates, it possesses a non-linear viscoelastic property [14].

PVB is mainly used as safety glass in laminated glass. With growing safety awareness and customer demand for safety features, laminated glass has become popular. Laminated glass is the formation of a sandwich-like structure of two or more layers of glass and a plastic interlayer. The purpose of the interlayer is to retain the fragments after a fracture has occurred, thus helping to reduce the risk of personal injury due to glass shards. It is often comprised of two glass plies bonded with an interlayer of plasticized PVB [15]. For example, in the automotive industry, laminated glass is used in windshields. In the aerospace industry, it is used in airplane windows, while for architectural purposes, it is used as glass panels for buildings. By using laminated glass, the safety, security, fire resistance, and sound attenuation properties can be enhanced [16]. Moreover, laminated glass can also resist high impacts such as bullets and blast loads, as well as being applicable for safety and security purposes such as protection against hurricanes and cyclones. These safety features will help to reduce personal injury and property damage during severe weather events [17]. In conclusion, laminated glass is a type of safety glass that holds together when shattered, since the interlayer (PVB) keeps the layers of glass bonded even when they are broken, and its high strength prevents the glass from breaking into large, sharp pieces. Virtually all laminated glass products produce the characteristics of the “spider web” cracking pattern when the impact is not enough to completely pierce the glass [18].

In this work, sugar palm fiber was employed as a reinforcement material in PVB polymer matrix to develop SPF-PVB laminated composites with different compositions through the hot compression method. The developed laminated composites were subjected to tensile properties and moisture absorption tests. The influence of different sizes of SPF and the composite ratio on the tensile strength, tensile modulus, tensile strain at maximum load, and moisture absorption were investigated.

## 2. Materials and Methods

### 2.1. Materials

#### 2.1.1. Sugar Palm Fiber

An image of the sugar palm fiber used in the current study is presented in Figure 1. The fiber used was sugar palm *(Arenga Pinnata)* fiber, which is in brown color, with the length of the sugar palm fiber being about 1.19m, and the fiber diameter ranges from 94 μm to 370 µm. The fiber is stiff and durable. The retting process was used to separate the stalk from the core part of the sugar fiber. The fibers were dried for about 2 weeks at room temperature after cleaning off the dirt. Then, these fibers were cut into two different sizes, i.e., short cut (0.1 cm–0.5 cm) and long cut (4.0 cm–5.0 cm). Sugar palm fibers have some advantages over traditional reinforcement fiber materials in terms of cost, renewability, non-toxicity, abrasiveness, density, and biodegradability [3].

#### 2.1.2. PVB Films

The PVB film was cut into a square shape with dimensions of 150 mm × 150 mm by using scissors. The polymer interlayer of PVB is tough and ductile, used for applications that require adhesion to many surfaces, strong bonding, flexibility, and toughness. Figure 2 shows the image of the PVB films.

### 2.2. Fabrication

Various compositions with different weight percentages of sugar palm fiber and PVB were produced by hot compression molding to form laminated composites. Firstly, certain layers of cut PVB film were placed on the mold, and the sugar palm fibers were placed in the middle, then other layers of PVB film were placed on top of SPF. The samples were then hot pressed with a load of 40 tons. The hot compression machine was set to a 165 °C temperature where the pre-heat time was 5 min, the heating time was 10 min, and the cooling time was 5 min. Thus, the total time taken in hot compression was 20 min. The laminated composite samples with dimensions of 150 mm long × 150 mm wide × 5 mm thick were fabricated, and the samples were then ready for testing. The image of sample laminate composites for short cut SPF is presented in Figure 3.

The three different compositions of the samples with different weight percentages of PVB and SPF were 80% PVB–20% SPF, 70% PVB-30% SPF, and 60% PVB–40% SPF. These sample were then differentiated by the size of SPF, either long cut (SPF-L) or short cut (SPF-S).

### 2.3. Characterization and Analysis

#### 2.3.1. Tensile Test

The speed of the Instron 3366 machine was fixed to a constant speed of 50 mm/min, and the sample was cut into a dog-bone structure with an overall length of 120 mm for the tensile test. The tensile strength is the maximal force per unit area acting on a plane transverse to the applied load and is a fundamental measure of the internal cohesive forces within a sample.

#### 2.3.2. Moisture Absorption Test

The moisture absorption text was performed at room temperature. The samples were cut into square shapes of 30 mm × 30 mm from the composite sheets. All samples were washed thoroughly and dried in an oven to remove all moisture inside the sample. The samples were then placed in a container filled with distilled water for different time intervals, which were 5 days, 10 days, 15 days, 20 days, and 25 days. The moisture absorption was determined by the mathematical formula given in Equation (1) [19]:(1)$$Moisture\;uptake\;\left(\%\right)=\frac{w_{f}-w_{0}}{w_{0}}\times 100$$
where
*w_f_* = mass of the sample at Day 25 (g)and *w*_0_ = mass of the sample at Day 1 (g)

## 3. Results and Discussion

### 3.1. Tensile Properties

#### 3.1.1. Modulus

Figure 4 presents the modulus of elasticity values for the composites. With the modulus being the measure of the stiffness of the samples, the trend in the graph shows that the higher the proportion of PVB, the higher the stiffness of the sample. Moreover, the different cuts of the sugar palm fiber also had a marginal influence on the modulus of the samples. This is because the dispersion of the fiber becomes worse with the increase in the fiber length. Therefore, the modulus of the short fiber is greater than the long fiber.

#### 3.1.2. Tensile Stress

The tensile stress value show the strength capacity of the samples in tensile loading. Figure 5 shows that the highest tensile stress was recorded for PVB 70-S followed by PVB 70-L, PVB 60-L, PVB 80-S, PVB 80-L, and PVB 60-S. The superior tensile stress exhibited by the PVB 70-S sample can be explained by (1) the higher concentration of SPF content (the breaking strength of fibers is much greater than that of the matrix), (2) the smaller size of SPF in PVB, which limits the stress concentration around the fiber during tensile loading relative to the counterpart sample with long SPF, and (3) the possible attainment of optimal wetting for SPF on PVB in this sample.

In general, the decrement in the tensile stress value of the different composites occurred beyond 30wt% SPF loading. This trend can be attributed to the influence of matrix discontinuity with rising filler content in the matrix, which resulted in poor stress transfer from the matrix to the filler [20]. Similar findings were reported elsewhere for thermoplastic starch/silk fiber composites [21], seaweed/polypropylene composites [20], pineapple leaf fiber/polypropylene composites [22], and sugar palm fiber/phenolic composites [23]. Meanwhile, the anomalous trend exhibited by the PVB 60 composites where PVB 60-S exhibited lesser tensile stress than the PVB 60-L counterpart can be attributed to the clustering of the short-sized fibers due to the high surface area of the particles and the high volume fraction of SPF in PVB, thus promoting fiber-fiber interactions and the subsequent defective strengthening of the composite. Moreover, a similar finding was reported by Facca et al. [24].

#### 3.1.3. Tensile Strain

Tensile strain is the measurement of the relative length of the deformation of the samples due to the tensile force applied. Figure 6 shows that the greater the proportion of PVB, the higher the tensile strain. Based on Figure 6, the different types of cut (long and short) influenced the tensile strain of the sample where the percentages of tensile strain for PVB 80-S, PVB 70-S, and PVB 60-S were all greater than the percentages of the tensile strain of PVB 80-L, PVB 70-L, and PVB 60-L. This shows that short cut SPF has greater tensile strain than the long cut SPF. The isotropic behavior is one of the reasons for this.

### 3.2. Moisture Absorption

For the development of reliable biocomposite materials, low moisture absorption is an essential factor to be considered in the selection of natural fibers as the reinforcing material since high absorption will adversely affect the dimensional stability of the composite, especially in terms of mechanical performance, porosity formation, and water retention capacity [25,26]. Figure 7 shows that PVB 60-L has the highest water uptake (and corresponding percent increment) compared to PVB 70-L and PVB 80-L, which are 0.068 g (2.32%), 0.031 g (1.04%), and 0.019 g (0.63%), respectively, after 25 days (water uptake was calculated by subtracting the weight of the sample after 25 days from the dry weight). This phenomenon can be attributed to the addition of different amounts of PVB and SPF. PVB has the properties of good water resistance, whereas SPF has poor water resistance. Hence, with the increasing proportion of PVB, the sample exhibited a lower amount of water uptake, whereas, as the proportion of SPF increased, the samples showed greater water uptake due to the properties of SPF. Longer fibers tend to increase the percentage of water absorption. The rate of water absorption is greatly affected by the specimen’s density and void volumes. The incorporation of long fibers into the mix decreased workability and increased the void space. Consequently, the longer the fiber, the higher is the water absorption. Hence, the introduction of compatibilizers, coupling agents, or chemical treatment can go a long way toward improving the moisture resistance of these composites [27].

## 4. Conclusions

In this work, the laminated composite sample with 80% of PVB and 20% of SPF showed the highest stiffness value. However, increasing the proportion of PVB too much results in high tensile strain. Furthermore, the comparison of two different cut sizes (long and short) was carried out, and their effect on the tensile test and moisture absorption were examined. For the tensile test, the modulus of the sample increased with a rising proportion of PVB due to the high bonding power and elasticity of PVB. Other than that, the short cut SPF exhibited a stronger modulus compared to the long cut SPF. For the moisture absorption test, the samples exhibited good resistance toward water with the increasing proportion of PVB due to its properties of good water resistance from, while SPF demonstrated poor water resistance. On the other hand, the long cut SPF showed better water resistance than the short cut SPF. From these results, the optimum composition for the best overall performance was PVB 80-S (20% SPF-80% PVB-short cut fiber), which is suitable for windshields and glass panels in buildings. Further surface treatment can be done to improve the water resisting ability.

## Figures and Tables

**Figure 1 polymers-12-01923-f001:**
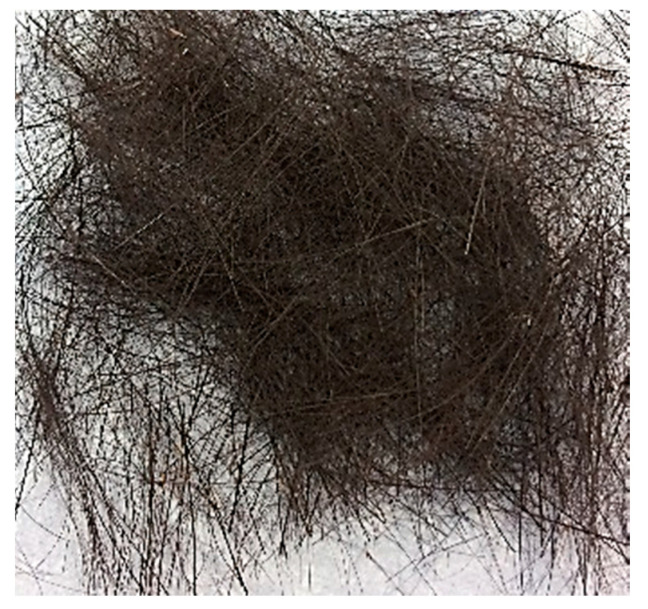
Sugar palm fiber.

**Figure 2 polymers-12-01923-f002:**
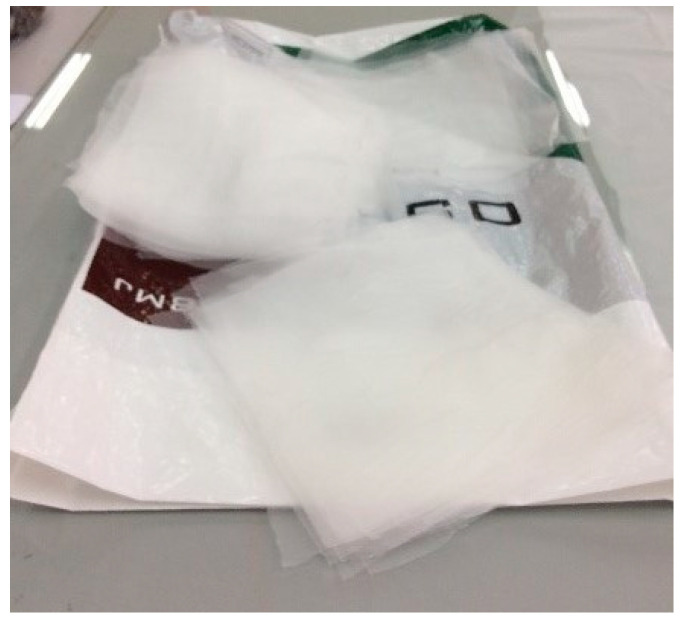
PVB films.

**Figure 3 polymers-12-01923-f003:**
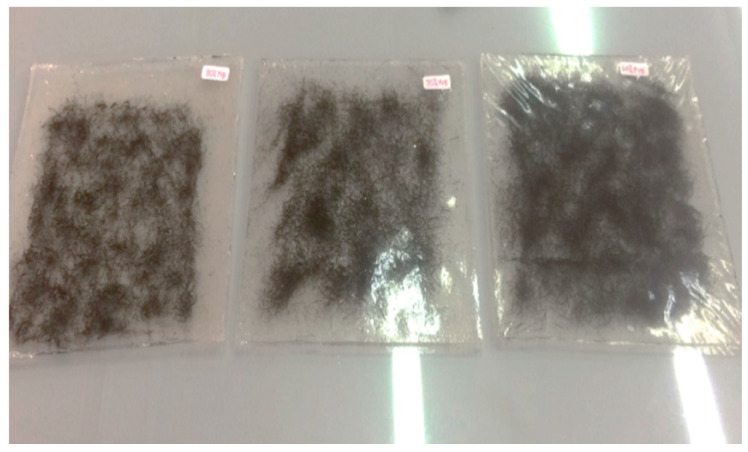
Samples of laminate’s composites for short cut SPF.

**Figure 4 polymers-12-01923-f004:**
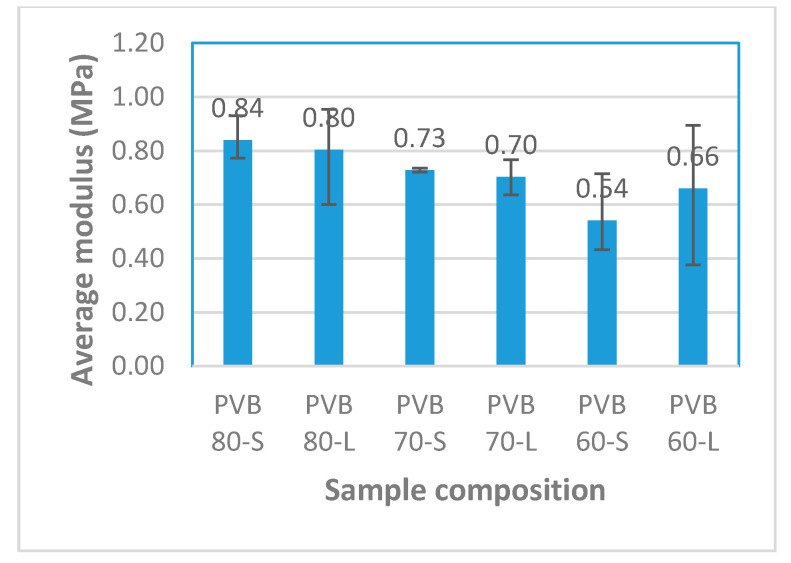
Average modulus of elasticity of the composite samples. S, short cut; L, long cut.

**Figure 5 polymers-12-01923-f005:**
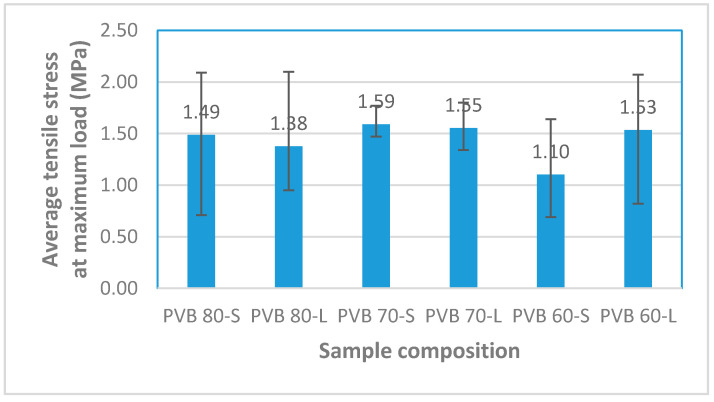
Average tensile stress at maximum load of the composite samples.

**Figure 6 polymers-12-01923-f006:**
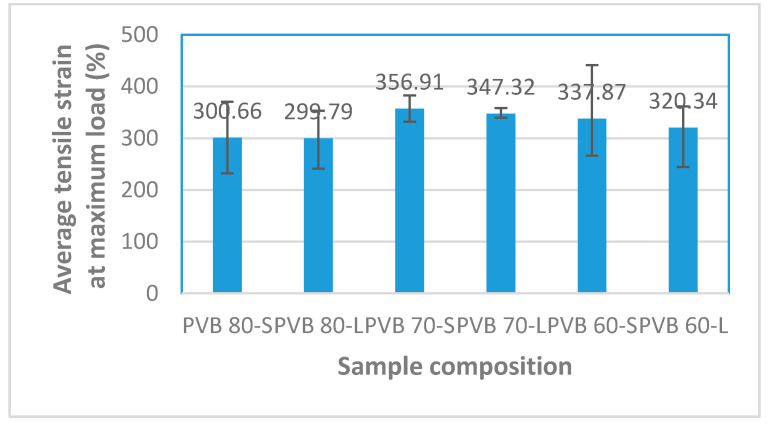
Average tensile strain at maximum load of the composite samples.

**Figure 7 polymers-12-01923-f007:**
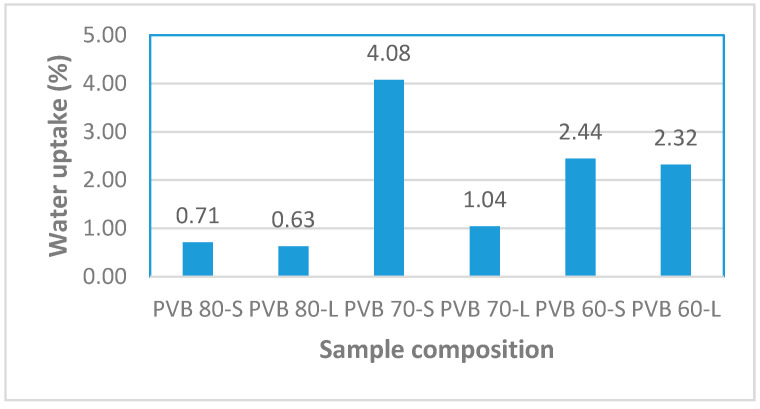
Percentage of water uptake of the composites after 25 days.

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
