# Peer review of "Tensile Strength and Moisture Absorption of Sugar Palm-Polyvinyl Butyral Laminated Composites"

_polymers, 2020, doi:10.3390/polym12091923_

Round 1

Reviewer 1 Report

This paper presented the fabrication of of Sugar Palm fiber reinforced polyvinyl butyral (PVB) laminated composites. The tensile properties and moisture absorption of the laminates were investigated.  The comments are summarized as follows:

1. Important experimental details were not provided in the paper, such as fiber content in the laminates, mechanical properties of the fiber, length of the fibers. How to disperse the fibers uniformly in the composites?

2. What was the potential application of the laminates? 

3. Figure caption for Fig 1 and error bars in Fig 2 were missing.

4. English needs to be polished. Quite a few mistypes or errors appeared.

Author Response

No.

Comments

Response

Page

1.

Important experimental details were not provided in the paper, such as fiber content in the laminates, mechanical properties of the fiber, length of the fibers. How to disperse the fibers uniformly in the composites?

Thank you for pointing this out. Authors have added the mechanical and other details of the fibers in section 2.

3

2.

What was the potential application of the laminates?

Authors have clarified the application details in introduction and conclusion.

9

3.

Figure caption for Fig 1 and error bars in Fig 2 were missing.

Authors have modified the figures.

4.

English needs to be polished. Quite a few mistypes or errors appeared.

Authors agree with the reviewer. We amended the manuscript.

Reviewer 2 Report

1) The methods are not adequately described.

2) The results are not clearly presented and need improvement.

3) The conclusion is not well written and needs modification.

4) The numbering system of the text is not appropriate.

5) Please explain why the short cut SPF exhibited stronger modulus compared to long cut SPF. why and What is the physical reason?

6)  The samples were differentiated by the size of the SPF either long cut or short cut. For the moisture absorption test, what is the difference between these two? It needs more discussion.  

Author Response

No.

Comments

Response

Page

1

The methods are not adequately described.

We appreciate the opportunity to add additional description about the methods involved.

3&4

2.

The results are not clearly presented and need improvement.

The results and discussion part has been amended.

6&7

3.

The conclusion is not well written and needs modification.

Authors have amended the conclusion part.

9

4.

The numbering system of the text is not appropriate.

Authors have corrected the numbering system throughout the paper

5.

Please explain why the short cut SPF exhibited stronger modulus compared to long cut SPF. Why and what is the physical reason?

We are grateful for this comment as it points to an important explanation.

6

6.

The samples were differentiated by the size of the SPF either long cut or short cut. For the moisture absorption test, what is the difference between these two? It needs more discussion.

Authors have clarified the differentiation and added more discussion on it in results and discussion part.

8

Reviewer 3 Report

The paper is generally well written however several things need to be commenetd/corrected, before publishing it in Polymers.

Comments/suggestions:

  1. Maybe the statement in lines 46-48 can be rephrased, since it is difficult to combine the values with the acronims.
  2. Some of the words are connected to each other - no spaces between. Maybe you can check it in the whole paper.
  3. Line 85: It should be number 2.
  4. Line 92 - is it wt. or vol. %?
  5. Line 94: There is no caption under the image. I uderstand that this is Figure 1?
  6. The symbol of degree C should be changed to the one from "symbols".
  7. Lines 95-97, between the number and "mintues" shoudl be a space.
  8. The quality of the equation 1 is really poor, maybe you can improve it, because now it is pixalized.
  9. Line 110- Results and discussion should be numbered as "3"
  10. Line 111 - Tensile Properties - point 3.1-instead of 1.1
  11. Figure 2. Lack of measurement error.
  12. Please provide stress-strain curves for the samples.
  13. line 127- "beyond"?
  14. Figure 3 - lack of measurement error.
  15. Figures 4 and 5- just as for gigure 2 and 4- lack of measurement error.
  16. Line 169 - Should it be point 3.2?
  17. The numbering in the paper, shoudl be corrected. Conclusions are also numbered as 1, it should be 4.

Author Response

No.

Comments

Response

Page

1.

Maybe the statement in lines 46-48 can be rephrased, since it is difficult to combine the values with the acronyms.

Authors have rephrased the lines by arranging values right next to the acronyms.

2

2.

Some of the words are connected to each other - no spaces between. Maybe you can check it in the whole paper.

Authors have amended the whole paper.

3.

Line 85: It should be number 2.

Authors have corrected the numbering system throughout the paper

4.

Line 92 - is it wt. or vol. %?

Authors have added the words “weight percentage” to that line

5

5.

Line 94: There is no caption under the image. I understand that this is Figure 1?

Whole Figures have been amended.

6.

The symbol of degree C should be changed to the one from "symbols".

Authors have changed the degree C to the one from symbols

5

7.

Lines 95-97, between the number and "minutes" should be a space.

Authors have modified the spacing error in those lines.

5

8.

The quality of the equation 1 is really poor, maybe you can improve it, because now it is pixalized.

It has been rewritten using proper Equation Editor.

9.

Line 110- Results and discussion should be numbered as "3"

Authors have corrected the numbering system throughout the paper

10.

Line 111 - Tensile Properties - point 3.1-instead of 1.1

Authors have corrected the numbering system throughout the paper

11.

Figure 2. Lack of measurement error.

Whole Figures have been amended.

12.

Please provide stress-strain curves for the samples.

The authors feel that it is not significant and the Young’s modulus is more important to be presented.

13.

line 127- "beyond"?

Thank you for pointing this typing error. We corrected it.

7

14.

Figure 3 - lack of measurement error.

Has been added

15.

Figures 4 and 5- just as for figure 2 and 4- lack of measurement error.

Has been added.

16.

Line 169 - Should it be point 3.2?

Authors have corrected the numbering system throughout the paper

17.

The numbering in the paper, should be corrected. Conclusions are also numbered as 1, it should be 4.

We have corrected the numbering system throughout the paper

Round 2

Reviewer 1 Report

It could be accepted after further polishing the english.